# The Interleukine-17 Cytokine Family: Role in Development and Progression of Spondyloarthritis, Current and Potential Therapeutic Inhibitors

**DOI:** 10.3390/biomedicines11051328

**Published:** 2023-04-30

**Authors:** Anna Davydova, Yuliya Kurochkina, Veronika Goncharova, Mariya Vorobyeva, Maksim Korolev

**Affiliations:** 1Research Institute of Clinical and Experimental Lymphology, Affiliated Branch of Federal Research Center of Cytology and Genetics, Siberian Division of the Russian Academy of Sciences, 630060 Novosibirsk, Russia; juli_k@bk.ru (Y.K.); varna21@mail.ru (V.G.); kormax@bk.ru (M.K.); 2Institute of Chemical Biology and Fundamental Medicine, Siberian Division of the Russian Academy of Sciences, 630090 Novosibirsk, Russia; maria.vorobjeva@gmail.com

**Keywords:** rheumatic diseases, IL-17A, aptamers, monoclonal antibodies, synthetic nucleic acids, aptamer therapeutics

## Abstract

Spondyloarthritis (SpA) encompasses a group of chronic inflammatory rheumatic diseases with a predilection for the spinal and sacroiliac joints, which include axial spondyloarthritis, psoriatic arthritis, reactive arthritis, arthritis associated with chronic inflammatory bowel disease, and undifferentiated spondyloarthritis. The prevalence of SpA in the population varies from 0.5 to 2%, most commonly affecting young people. Spondyloarthritis pathogenesis is related to the hyperproduction of proinflammatory cytokines (TNFα, IL-17A, IL-23, etc.). IL-17A plays a key role in the pathogenesis of spondyloarthritis (inflammation maintenance, syndesmophites formation and radiographic progression, enthesites and anterior uveitis development, etc.). Targeted anti-IL17 therapies have established themselves as the most efficient therapies in SpA treatment. The present review summarizes literature data on the role of the IL-17 family in the pathogenesis of SpA and analyzes existing therapeutic strategies for IL-17 suppression with monoclonal antibodies and Janus kinase inhibitors. We also consider alternative targeted strategies, such as the use of other small-molecule inhibitors, therapeutic nucleic acids, or affibodies. We discuss advantages and pitfalls of these approaches and the future prospects of each method.

## 1. Introduction

Spondyloarthritis (SpA) represents a group of chronic inflammatory rheumatic diseases with a predilection for the axial skeleton and sacroiliac joints, also characterized by the development of arthritis, enthesitis, dactylitis, and extramusculoskeletal manifestations (uveitis, psoriasis, and inflammatory bowel disease) [1].

The group of spondyloarthritis includes axial spondyloarthritis, psoriatic arthritis, reactive arthritis, arthritis associated with chronic inflammatory bowel disease (ulcerative colitis and Crohn’s disease), and undifferentiated spondyloarthritis. The axial spondyloarthritis subgroup is, to date, divided into radiographic axial spondyloarthritis (rx-axSpA) and non-radiographic axial spondyloarthritis (nr-axSpA), characterized by the absence of sacroiliitis on X-ray [2]. The prevalence of SpA in the population varies from 0.5 to 2% [3], most commonly affecting young people [4].

Radiographic axial spondyloarthritis is widespread in the population (9–30 cases per 10,000 population) and develops in individuals under 40 years of age [5]; somewhat less common is psoriatic arthritis (8–17 cases per 10,000 population) [6]. The incidence of spondyloarthritis in patients with inflammatory bowel disease ranges from 4% to 20% [7], and its prevalence is also higher at a young age: from 20 to 30 years [8].

Features of the clinical manifestations of diseases from the SpA group and their predominant development in working-age persons bring significant economic burden and increased use of health care resources; meanwhile, achieving low activity or remission is associated with a decrease in both direct costs (hospitalizations, emergency services, and unscheduled doctor visits) [9], and indirect costs (temporary and permanent disability) [10]. Patients with active spondyloarthritis have high rates of general hospitalization for various reasons, including emergency room visits, outpatient visits, and investigations [11]. Patients with psoriatic arthritis have more frequent and longer periods of temporary disability compared to their peers [12]; up to 74.1% of patients with active axial spondyloarthritis face employment problems [13], and almost a quarter (23.4%) are unable to work fully due to illness [14]. Persistent disability, according to various sources, occurs in 13–45% of patients with axial spondyloarthritis [4], and in 16–39% in psoriatic arthritis patients [15].

One of the severe manifestations of axSpA which significantly worsens the prognosis is the involvement of the hip joints in the form of coxitis. The defeat of the hip joints in axSpA occurs in 56% of cases [16], and 5% of patients require a total arthroplasty [17]. For psoriatic arthritis, the involvement of the hip joint reaches 34% [18], and 1.8% of patients need surgical treatment [19]. There is also a high risk of developing aseptic necrosis of the hip joint before the age of 30 [20]. The average age of patients undergoing hip arthroplasty is 48 years [21]. Biological disease-modifying antirheumatic drugs (bDMARDs) enable the reduction in the need for a total hip arthroplasty [22]. The development of osteoporosis is also considered as a complication of spondyloarthritis, which leads to a higher risk of fractures with subsequent disability, compared to the general population [23,24].

The development of the complications described above, a high level of disability, and an increase in healthcare system costs are associated with late diagnosis of the disease and failure to control it [25]. Even a 6-month delay in starting therapy is associated with worse functional outcomes [26]. However, the mean delay in diagnosis is currently 5.7 years, and the presence of psoriasis is highlighted as a factor associated with later diagnosis [27]. Delayed diagnosis of spondyloarthritis leads to a late start in therapy, decreases its efficacy, and significantly elevates the level of disability. The latter issue is critical, since the disease develops at a young working age [28]. At the same time, timely diagnosis and active treatment with the achievement of control over the disease retains patient performance and is economically favorable from the point-of-view of the public health system [29].

SpA pathogenesis is related to hyperproduction of proinflammatory cytokines (TNFα, IL-1β, IL-6, IL-8, IL-17A, etc.). More evidence is received about the role of IL-17A in SpA pathogenesis: inflammation maintenance, syndesmophites formation and radiographic progression, enthesites and anterior uveitis development, etc. [30,31].

Delayed diagnostics and inadequate treatment bring such complications as secondary osteoarthritis, osteoporosis with pathological fractures, aortic valve lesions, and amyloidosis.

The management of patients with axSpA includes non-pharmacological and pharmacological interventions. To achieve disease control, ASAS/EULAR recommends non-steroidal anti-inflammatory drugs, tumor necrosis factor inhibitors, interleukin-17 inhibitors, and Janus kinase inhibitors [32].

According to clinical management recommendations, inhibitors of eighter TNFα or IL-17A are the first-line biologic treatment. It has been proven that therapy with IL-17A inhibitors for at least two years influences the radiographic progression in contrast to TNFα inhibitors [3]. To date, three IL-17A-inhibitors have been registered: sekukinumab, ixekizumab, and netakimab. However, for all their benefits, such as high clinical efficacy and an acceptable safety profile for the patient, therapeutic anti-IL-17A antibodies also possess a number of limitations: immunogenicity, difficulties in transportation and storage, and high cost [33,34]. In recent years, other targeted molecules have been proposed for the purpose, such as therapeutic nucleic acids or nanoantibodies (see Figure 1).

In this review, we consider the role of IL-17 family interleukins in the pathogenesis of SpA, analyze the literature on the recent strategies for affecting the level of IL-17 cytokines, and discuss the merits and demerits of different approaches.

## 2. IL23/IL17 Axis in the Pathogenesis of Autoimmune Inflammation

The IL23/IL-17 axis plays a key role for IL-17-mediated pathological effects. This signaling pathway includes IL-23-induced IL-17 production by Th17 cells due to activation of the JAK-STAT cascade and induction of the transcription factor RORγt. The physiological function of this axis provides antibacterial and antifungal effects. Otherwise, pathological function of the axis is determined by the IL-23 contribution to the formation of the inflammatory phenotype of Th17 cells, characterized by IL-17 hyperproduction. Aside from their own pathological effects, interleukins-17 also induce an activation of several cytokine cascades, resulting in the expression of GM-CSF, CCL, IL-1, IL-6, and TNFα (Figure 2) [35,36,37].

IL-23 is a proinflammatory cytokine mainly secreted by activated macrophages and dendritic cells. Its receptor (IL-23R) is expressed on T-cells, innate lymphoid cells, intraepithelial lymphocytes, natural killer cells, intestinal epithelial cells, and granulocytes. Stimulation of IL-23R activates the signaling cascade JAK-STAT, resulting in proinflammatory cytokine production [35]. The differentiation of Th17 cells from naïve Th0 starts by TGFβ, IL-6, and IL-21, with further activation of transcription factors and IL-23R expression. Then, IL-23 induces the stabilization of the key transcription factor RORγt, along with final differentiation and proliferation of the Th17 phenotype with high levels of expression of T-bet, IL-23R и GM-CSF. The latter is pivotal for the active production of the pro-inflammatory cytokines, including IL-17, suppression of pro-inflammatory cytokine IL-10, and the implementation of pathological effects [36,37]. IL-23 enhances and maintains the TGFβ expression, thus indirectly mediating an initiation in Th0 to Th17 differentiation. Moreover, the IL-23-induced expression of IL-23R gains the pathogenic effect by the feedback mechanism [38]. Otherwise, numerous effects of IL-17A and IL-17F include an ability to influence a wide range of cells and induce an expression of pro-inflammatory cytokines, especially the IL-6 necessary for Th17 differentiation [37]. In the absence of the interleukin-23, IL-6, TGFβ, and IL-21 stimulate the differentiation of Th0 to the IL-10-producing Th17 cells that constrain pathogenic reactions [39].

Th17 cells have a high level of flexibility which allow a co-expression of alternative, lineage-determining transcription factors and cytokines [39]. Therefore, Th17 can transform into different cell subsets, including Tregs, Th1 cells, Tr1, and follicular T-cells. Moreover, Th17 cells are capable of the transient co-expression of RORγt with FOXP3, and IL-17F with IL-10. In vitro studies have shown that TGF-β1 induces Th17 transformation to Tr1 by means of the SMAD3 pathway, while the joint action of TGF-β1, IL-6, and IL-23 mediates the development of potentially pathogenic Th17 cells [40]. In the absence of IL-23, TGF-β1 and PGE2 induce the transformation of Th17 to Treg FOXP3+, which cannot produce IL-17 [41]. Vice versa, in the presence of IL-6 and IL-23, Treg FOXP3+ acquire the Th17-like phenotype [42]. The cells with a simultaneous co-expression of FOXP3+ and RORγt+ have been also described; such T-cell precursors express both transcription factors until expression of one predominates to generate Tregs or Th17 cells. Presumably, such T-cells acquire the characteristics of Th17 under the action of IL-23, which enhances RORγt expression, or Treg FOXP3+ characteristics, in the absence of IL-23 [43]. Cell and animal model studies have also revealed an ability of IL-23 mediated by RORγt-signaling to induce the accumulation of FOXP3+ Tregs with elevated proliferation, co-expression of RORγt, and production of IL-17A [44].

Therefore, the balance between the activities of IL-23 and IL-17, together with the flexibility of Th17 cells, determines a possibility of pathogenic autoimmune inflammatory responses, in particular, the induction of spondyloarthritis.

## 3. IL-17 Family in the Pathogenesis of Rheumatic Diseases

As we have briefly mentioned in the Introduction, the SpA group includes five main nosologies, which will be described below in more detail. All of them possess common pathogenic and clinical features. In particular, the common feature for all SpA is a spine joint lesion that is characterized by inflammatory pain and stiffness. Moreover, other SpA features include peripheral arthritis, enthesites, uveitis, dactylitis, and gastrointestinal tract involvement, wherein the symptom severity has its own traits depending on the specific nosology. IL-17 family members play a critical role in the pathogenesis of the nosologies united into the SpA group. The physiological and pathological effects of IL-17 family members are summarized in Table 1.

The IL-17 family consists of six structurally related cytokines: IL-17A (IL-17), IL-17B, IL-17C, IL-17D, IL-17E (IL-25), and IL-17F (Table 1). Among them, the most studied members are IL-17A, IL-17B and IL-17E. The IL-17 family plays an important role in antifungal and antibacterial protection, regulation of skin, oral, and intestine microbiota, and promotes tissue and skin repair. Recently, IL17B has attracted the attention of researchers due to its expression in various tissues. This cytokine participates in tumor progression and presumably has effects on the development of cartilage and bone tissues. An overexpression of IL17B has been determined in fractures and other injuries of the musculoskeletal system. An activation of the IL17B receptor was demonstrated during induced inflammation [65].

Interleukin IL17E, also known as a barrier cytokine, is associated with skin, respiratory, and gastrointestinal tracts. The IL17E role in psoriasis development has been reported; nevertheless, there is no convincing evidence for its role in the development of articular syndrome [89]. It is believed that IL17F may play a role in the development of psoriasis and psoriatic arthritis, but its effects are not as pronounced as those of IL17A [15].

*Axial spondyloarthritis (axSpA)* is a chronic axial disease which manifests with inflammatory pain rhythm and spinal stiffness. Axial SpA includes axial radiographic spondyloarthritis with a radiographic sacroiliitis as an obligatory criterium [95]. In 2009, the European-based Assessment of SpondyloArthritis International Society (ASAS) developed and published non-radiographic axSpA criteria that allow the establishment of a diagnosis for patients who have inflammatory back pain without radiographic sacroiliitis [96,97]. Non-radiographic axSpA is characterized by MRI signs of inflammation in the sacroiliac joints, which include, according to the consensus of the ASAS/OMERACT MRI group, bone marrow edema (osteitis), synovitis, enthesitis, and capsulitis [98]. It should be noted that bone marrow edema is a key criterion for a sacroiliitis diagnosis. Moreover, the ASAS 2009 criteria allow for verifying of the axSpA diagnosis even without the sacroiliac joint changes, based on HLA-B27 presence together with two or more characteristic clinical manifestations (inflammatory back pain, arthritis, enthesitis, uveitis, dactylitis, psoriasis, IBD, efficacy of NSAIDs, family history of spondyloarthritis, and an elevated C-reactive protein) [96,97]. The main clinical features are spondylitis, arthritis, uveitis, dactylitis, and an elevated level of C-reactive protein. Nowadays, axSpA etiology remains poorly understood, while its pathogenesis is being actively investigated. It is considered that about 90% of patients with SpA carry HLA-B27, but only 1–2% of HLA-B27-positive people develop the disease. According to one of the theories, HLA-B27 presents arthritogenic peptides to CD8+ T-lymphocytes. Interaction of the T-lymphocytes with the HLA-B27 peptide complex results in their activation and maintains an autoimmune inflammation [99]. Otherwise, a microbial mimicry hypothesis supposes that some microbial antigens are structured similar to autoantigens and thus can activate CD8+ T-lymphocytes via interaction with the HLA-B27. HLA-B27 heavy chains can form dimeric complexes linked by disulfide bonds, which have been found in the gut mucosa and synovial tissue of SpA patients. These complexes are also present on the surface of dendrite cells and force T-lymphocytes to produce IL-17 [100]. During the antigen presentation under proinflammatory cytokine influence (Figure 2), Th0 differentiate in Th17, which are responsible for IL-17 production.

The role of IL-17A is most thoroughly investigated in the axSpA pathogenesis. Key axSpA pathogenic features include the formation of erosions and new bone tissue growth, and the role of IL-17A in osteoresorption and osteoproliferation has been established. It is considered [101,102] that IL-17A stimulates expression of the receptor–activator nuclear factor-kB ligand (RANKL) and inhibits Wnt signaling, thus suppressing osteoblast activity, promoting osteoclast differentiation and bone resorption. Simultaneously, mesenchymal stem cells intensely differentiate into osteoblasts [103], in particular through JAK2/STAT3 activation [30], and their active proliferation results in new bone formation in the form of syndesmophyte. IL-17A also participates in the development of enthesites. Its hyperproduction within the entheses is launched by PgE2 and IL-23 and is linked with resident T-cell activation [104]. Arthritis development in axSpA is induced by naive T-cell differentiation into Th17, γδT, and ILC3, stimulated by proinflammatory cytokines (TGFβ, IL-6). These cells produce a lot of IL-17, whose effector mechanisms include the activation of fibroblasts and endothelial cells, as well as RANKL activation. This cascade of events results in the formation of erosive arthritis. While TNFα is generally thought to participate in the pathogenesis of uveitis, and anti-TNF drugs are effective for panuveitis and posterior uveitis treatment, IL-17A and IL-17F are also reported to be involved in anterior uveitis progression [105].

*Psoriatic arthritis (PsA)* is a disease of the SpA group which is associated with skin psoriasis. Its clinical features are characteristic for SpA and include dactylites, enthesites, arthritis, spondylitis, uveitis, and intestinal manifestations. Genetic risk factors play a certain role in disease development, and some HLA alleles are associated with PsA progression. For example, B*08, B*27, and B*38 can induce PsA development, whereas HLA-B*27:05:02 is associated with arthritis, enthesites, dactylites, and symmetric sacroiliitis, and HLA-B*08:01:01 and HLA-C*07:01:01 are associated with erosive arthritis, dactylites, and asymmetric sacroiliitis [106,107]. PsA development is thought to be connected to prolonged chronic skin inflammation during psoriasis. Injury-affected keratinocytes secrete antimicrobial peptides that specifically stimulate plasmocytoid DCs for INF I production which, in turn, lead to myeloid DC maturation [108]. The migration of DC to regional lymph nodes, where immune response is proceeding, induces T-cell activation, IL-23 production, and activation of Th17 that produce high levels of IL-17 and maintain inflammation cascade [109].

The development of enthesites is one of the key PsA symptoms and sometimes gets ahead of arthritis. Mechanical stress in entheses insertions causes immune cell activation (Th17, γδT, MAIT, ILC3) and hyperproduction of IL-23 and IL-17A. It has been shown that a subpopulation of γδT-cells is able to produce IL-17 regardless of IL-23 expression [110]. A series of studies demonstrate the effectiveness of IL-17A blocking in PsA enthesites [111]. Moreover, targeted blocking of the IL23/IL17 axis has also shown clinical efficacy in regard to enthesites in PsA [112]. The mechanism of erosions and syndesmophyte formation in PsA, as in axSpA, is based on the IL-17A effects on osteoblasts, macrophages, fibroblasts, and endothelial cells that produce proinflammatory cytokines and influence osteoclast activation through RANKL.

*Spondyloarthritis associated with Inflammatory Bowel Diseases (IBD)*. IBD is a group of chronic inflammatory bowel diseases, which includes Crohn’s disease, ulcerative colitis, and non-classified IBD. Nowadays, a definite relationship has been established between IBD and radiological sacroiliitis, as well as SpA [113]. It has been shown that IBD patients develop SpA more often as compared to those without IBD [114]. Clinical, genetic, and immunological relationships have been found between SpA and IBD [113]. Genome association studies have revealed common loci associated with both SpA and IBD progression, including associative signals in the genes of the IL12/23 pathways [115,116].

The gut microbiota, which under normal conditions maintains mucosa integrity and has an anti-inflammatory effect, plays a great role in IBD progression. Studies on mouse arthritis models have shown that changing of the gut microbiota by interaction with DCs leads to Th17 activation [117]. The presence of *Chlamydia trachomatis* induces IL-23 expression and IL-17 production [118]. Moreover, elevated TNFα expression by epithelial cells due to the microbiota’s activity provides immune cell recruitment and mucosal injury, thus maintaining chronic inflammation. A therapy by TNFα inhibitors has shown a good clinical response in IBD, and some authors have reported on the normalization of the gut microbiota during such therapy [119,120]. Despite the proven role of IL-17 in IBD and SpA pathogenesis, a clinical trial has shown that, contrary to expectations, anti-IL-17 therapy by Sekukinumab aggravated the course of Crohn’s disease, so the trial has been stopped [121]. The use of brodalumab, an IL-17 receptor antagonist, gave similar results [122]. It is considered that IL-17 inhibition worsens the course of Crohn’s disease and can cause its progression. There is a hypothesis that IL-17 plays a role in gastrointestinal homeostasis and possibly protects the gut mucosa, in contrast to its proinflammatory role in SpA. IL-17A blockage can disbalance and dysregulate mucosal cytokines and induce Th1 activation, which results in gut mucosa inflammation and promotes IBD [123]. Another hypothesis states that IL-17 inhibition changes gut microbiota composition via the excessive growth of *Candida albicans* that leads to IBD development in susceptible people [124]. Herewith, blockage of IL12-/23 axis has shown clinical efficacy for induction of remission and maintenance of low activity of Crohn’s disease and ulcerative colitis [125,126].

*Reactive arthritis (ReA)* also relates to the spondyloarthritis group. It is associated with T-cell activation caused by urogenital or gastrointestinal infection. The disease usually starts in the age range from 18 to 40 years old and does not have a certain gender preference. The pathogenesis of reactive arthritis is now considered to be associated with environmental and genetic factors. Special attention is given to infectious agents such as chlamydia, salmonella, and yersinia [127]. Bacterial peptides and cell wall components from infectious focus get into synovia, interact with antigen presenting cells, and cause an immune reaction, including activation of T-cells and macrophages, production of proinflammatory cytokines, and development of synovitis. In this context, *Cl. Trachomatis* plays a special role due to its ability to inhibit the merging of lysosomes and phagosomes and to persist intracellularly. Salmonellae cause an activation of γδT-cells that produce IL-17 and promote arthritis, spondylitis, and conjunctivitis [128].

Genetic risk factors are associated with HLA-B27 carrying. Anywhere from 50 to 80% of patients with reactive arthritis are HLA-B27 positive. HLA-B27 has several alleles that can influence susceptibility to disease development. For example, HLA-B*2703 increases the risk of a typical ReA clinical triad [129]. It has also been reported that HLA-B27 contributes to maintaining some bacteria in a host organism, especially Chlamydia and Salmonella [130]. Therefore, genetically predisposed people respond to infectious agents by intensified Th17 differentiation, elevated IL-17 production, and by maintaining inflammation. Herewith, an activity of NK-cells allows for the decreasing of IL-17 production by γδT-cells, while gut lactobacilli can inhibit the expression of IL-17, IL-23, and TNFα and play a protective role during ReA progression [131].

*Undifferentiated spondyloarthritis (uSpA).* Undifferentiated SpA is a state that possesses some typical SpA clinical features, but fails to meet certain criteria for AS, PsA, SpA associated with IBD, etc. It is supposed that uSpA could be an early stage of rx-axSpA. Otherwise, an issue is discussed that uSpA is a separate nosology which may not transform into another one [132]. Qing Xia et al. [133] in a meta-analysis have shown that nearly 22% of patients with uSpA undergo transformation into AS after 5 years, and 29,1% and 39,9% of patients after 8 and 10 years, respectively. Disease pathogenesis is common for all the SpA group with the key role of HLA-B27 carrying and Th17 activation that leads to arthritis, enthesitis, and spondylitis due to IL-17 hyperproduction.

*Juvenile idiopathic arthritis (JIA)* includes all forms of chronic arthritis of an unknown etiology, which debuts before the age of 16 and persists for at least 6 weeks. JIA is subdivided into several forms: oligoarticular, polyarticular, JIA with systemic onset, psoriatic arthritis, enthesitis-associated, and undifferentiated JIA. Axial involvement of the sacroiliac joints, spine, and association with HLA-B27 is characteristic of enthesitis-associated JIA, while juvenile psoriatic arthritis represents an equivalent form of psoriatic arthritis [134]. JIA is characterized by an increased IL-17 level and Th17 cells in the blood and synovial fluid, as well as a predominance of Th17 cells with RORC expression over the Treg FOXP3+ population. A direct inhibition of IL-17 by monoclonal antibodies has shown positive clinical effects for JIA patients [135].

Therefore, the described nosologies fall into the same SpA group due to similar clinical features and common stages of pathogenesis. The main role belongs to IL-17 hyperproduction in response to immune cell activation by various factors, and so blockage of IL-17A or IL-17 receptors provides prominent clinical effects. While inhibitors of TNFα and Janus kinases [32] are effectively used for SpA therapy, only IL-17A inhibitors are proven for the slowing of radiographic progression and structural changes [136,137,138].

## 4. Selective Inhibition of IL17 Cytokine

### 4.1. Monoclonal Antibodies

Inhibition of the IL-17 axis by monoclonal antibodies is the most thoroughly investigated approach in terms of clinical efficiency and patient safety. Therapeutic antibodies suppressing the IL-17/IL-17R axis will be reviewed in the next subsection.

Nowadays, the most prevalent technique for precise inhibition of cytokines is the use of monoclonal antibodies that specifically bind the cytokine itself, or its receptor. The first monoclonal antibodies were opened by Milstein and Köhler (Cambridge University) [139] and represented mice proteins, therefore high immunogenicity limited their applications in clinical practice. Further, the technology of replacing the mice fragments of antibodies with human ones (chimerization) allowed for lower immunogenicity, and humanization technology (transfer of hypervariable mice fragments onto human IgG) and was the next step to solve the problem. Currently, technologies have been developed for producing fully human antibodies [140].

The first clinical trials of the efficacy and safety of anti-IL-17 mAb in humans were performed on rheumatoid arthritis patients with humanized mAb anti-IL-17-LY2439821 [141], also known as ixekizumab. However, anti-IL-17 therapeutic antibodies have not shown the expected effect on rheumatoid arthritis, and currently are not used in this nosology.

Secukinumab, a completely human IgG antibody binding IL-17A (Table 2) became the first anti-IL-17 therapeutic mAb approved for use in humans.

According to the ERASURE (738 patients) and FIXTURE (1306 participants) clinical trials, secukinumab demonstrated efficacy on psoriasis by decreasing skin and nails lesions and skin itch, and improving life quality [142]. Secukinumab also showed an effect on PsA activity in the FUTURE 1 (606 participants) [143] and FUTURE 2 (397 patients) [144] trials, having an impact on arthritis, dactylitis, enthesitis, and slowing radiographic progression. The MEASURE 1 (371 patients) and MEASURE 2 (219 patients) trials demonstrated high secukinumab efficacy in AS: 61% of patients on subcutaneous injections of 150 mg showed an ASAS20 response by the 16th week, and clinical effects were retained for 52 weeks of treatment [145]. Secukinumab have also been registered for non-radiographic axSpA treatment after the PREVENT trial [146]. There was an ASAS40 response in 42.2% of patients by the 16th week, a clinically significant decrease in disease activity as compared with the placebo, and statistically significant improvement of spine mobility and life quality.

A study by Pavelka et al. showed the achievement of an ASAS20 response in more than half of the observed cases of secukinumab treatment of active rx-axSpA by the end of the 16 weeks [147]. An ASAS20 response was achieved by 66% of participants by the 12th week. The positive effect of secukinumab on AS pathology development was proved by the slowing of radiographic progression [148]. After 24 weeks of secukinumab treatment, patients with psoriatic spondyloarthritis showed a decrease in synovitis, as well as the absence of progression of catabolic and anabolic changes in the bones [149]. The EXCEED study compared the efficacy of secukinumab and a TNF inhibitor adalimumab, and demonstrated their equal efficacy and safety [150]. Another SURPASS study involved 858 patients with active rx-axSpA characterized by the presence of syndesmophytes in the cervical and/or lumbar part of the spine, and sacroiliitis, treated with secukinumab or adalimumab [151]. It was shown that IL-17A inhibition is more effective as compared to TNFα inhibition in terms of the radiological progression of spondyloarthritis. At the same time, a study by Khatri et al. [152] included 221 patients with rheumatoid arthritis and 240 patients with psoriatic arthritis. Simultaneous neutralization of both TNFα and IL-17A showed no effect on the additional blockade of IL-17A in the presence of TNFα inhibition. However, high dosages of the IL-17A inhibitor provided a better clinical response. Genovese et al. [153] obtained similar results for the double inhibition of TNFα and IL-17 in patients with rheumatoid arthritis.

A prospective IVEPSA study showed that IL-17 inhibition by secukinumab in psoriasis patients with a high risk of arthritis could prevent the development of articular pathology [154]. Suppression of IL-17 also led to the abruption of osteoproliferation, the absence of enthesiophyte progression, and a reduction in periarticular inflammation. During the study, none of the participants were diagnosed with spondylitis.

Ixekizumab is a humanized anti-IL-17A mAb which inhibits the IL-17A interaction with its receptor [155]. It was registered in the USA and EU in 2016, and in the Russian Federation in 2018. Ixekizumab is approved for use in moderate-to-severe and severe plaque psoriasis in adults and children from 12 years of age, and in active PsA, rx-axSpA, and nr- axSpA in adults.

The efficacy of ixekizumab against plaque psoriasis has been demonstrated in three placebo-controlled studies: UNCOVER-1 (1296 patients), UNCOVER-2 (1224 patients) and UNCOVER-3 (1346 patients) [156]. The efficacy and safety of ixekizumab use in PsA have been shown in the randomized double-blind placebo-controlled trials SPIRIT-P1 and SPIRIT-P2. The effect on musculoskeletal symptoms, decrease in skin lesions, and slowing of PsA structural progression have been shown [155].

The results of the COAST-V and COAST-W trials in axSpA show decreased disease activity by the 16th week of treatment with maintenance of the effect over 52 weeks, improvement of functional changes according to BASFI, and decrease in inflammatory activity in the spine and sacroiliac joints according to MRI [157]. A randomized double-blind placebo-controlled study demonstrated the proved efficacy of ixekizumab in a group of 303 patients with nr-axSpA as compared to the placebo [158]. Another clinical study by Dougados et al. [157] approved successful use of ixekizumab for the treatment of 341 patients with rx-axSpA. The suppression of radiographic progression in more than half of the patients was shown over the course of the 156 weeks of the study [159].

Netakimab is a recombinant humanized mAb specifically binding IL-17A produced by BIOCAD, Russia [138]. It was registered in Russia under the tradename Efleira in 2019. Netakimab (Efleira) has been approved for use in rx-axSpA, psoriatic arthritis, and psoriasis in the Russian Federation and Belarus, but has not yet been approved by the FDA and EMA. Registered indications of Efleira include treatment of moderate-to-severe and severe plaque psoriasis, treatment of axSpA, and treatment of active PsA in adults. The efficacy and safety of using netakimab in plaque psoriasis have been shown in the PLANETA trial (213 patients) [160]. PASI75 response by the 12th week was reached by 77.7% of the patients on a 120 mg dosage every 2 weeks, and by 83.3% of the patients every 4 weeks, and the clinical effect was retained over the course of a year. Moreover, netakimab showed a low percent of side effects.

The results of netakimab use in PsA were shown in the PATERA trial (194 patients) [161,162]. The ACR20 response was reached by 82,5% of patients, accompanied by the mitigation of clinical features and a statistically significant decrease in ASDAS-CRP and BASDAI indexes.

The efficacy of netakimab in axSpA has been evaluated in the BCD-085-5/ASTERA trial (228 patients) [163]. Netakimab-treated patients demonstrated an authentic decrease in CRP, improvement of functional ability by BASFI, and slowing of radiographic progression by mSASSS.

Bimekizumab is an IgG monoclonal antibody specifically binding both IL-17A and IL-17F [164]. It was registered in the EU in 2021 under the tradename Bimzelx (UCB Pharma SA, Belgium) and approved for use in moderate and severe psoriasis in adults. The possibility of using Bimekizumab for PsA and axSpA treatment is now under investigation. Patients treated with bimekizumab had higher combined response rates for ACR20, ACR50, and ACR70 compared with a placebo after two weeks [93]. The results of the BE ACTIVE trial (206 participants) [165], revealed a considerable, stable decrease in pain intensity and improvement in life quality in patients with active PsA. In the BE AGILE trial (303 patients) [166], ASAS40 response was reached by 45.9% of the patients with active axSpA on 320 mg, and by 46.7% of patients on 160 mg of bimekizumab. The BE MOBILE 2 trial (332 participants, 160 mg of bimekizumab every 4 weeks) [167] significantly decreased CRP levels by the 2nd week of treatment, reduced an active inflammation in sacroiliac joints and spine according to MRI by the 16th week, and by the 24th week more than 50% of the patients reached an ASDAS < 2.1. Therefore, bimekizumab represents a promising, effective, and safe drug for PsA and axSpA treatment.

Brodalumab is an IgG2 mAb specific to the IL-17A receptor, which inhibit the effects of IL-17A, IL-17F, IL-17C, and IL-25 [168]. It has been approved for the treatment of moderate-to-severe and severe psoriasis in adults in the EU as Kyntheum (LEO Pharma, Ballerup, Denmark), and as Siliq in the USA (Valeant Pharmaceuticals, Laval, QC, Canada) in 2017. Brodalumab, as Lumicef (Kyowa Hakko Kirin, Tokyo, Japan), was approved in 2016 in Japan for the treatment of psoriasis vulgaris, PsA, pustular psoriasis, and erythrodermic psoriasis [169].

Brodalumab showed the efficacy and safety of the treatment of plaque psoriasis in adults in the phase III trial AMAGINE [170] with improvement in the state of skin and nails, significant decrease in the affected area (PASI75), and improvement of life quality (HR-QOL); the effect lasted for 52 weeks. Randomized phase III trials AMVISION-1 and AMVISION-2 [171] investigated the possibility of using brodalumab in PsA; a ACR20 response by the 16th week was reached by 45.8% patients on 140 mg, and by 47.9% patients on 210 mg, together with the mitigation of dactylites and enthesites.

Some therapeutic anti-IL-17 mAbs are now undergoing the early stages of investigation. For example, Min Wu et al. have published the results of the phase I clinical trials of QX002N [172]. QX002N is a humanized IgG1 anti-IL-17A mAb. According to the authors, the drug has shown a good safety profile, and further investigations are now ongoing.

Nanobodies represent a novel class of focused antibody-like therapeutics made of camelid-derived single-domain antibodies [173]. They combine the merits of small proteins, such as higher stability and less complex production, with the affinity and specificity of monoclonal antibodies. The application of nanobodies for autoimmune disease treatment is now being extensively investigated. In particular, the nanobody sonelokimab inhibition of both IL-17A and IL-17F is under clinical study for its efficacy and safety in patients with psoriasis [174]. Another clinical study that included 44 patients with psoriasis demonstrated a decrease in inflammatory activity during sonelokimab treatment [175].

The main side effects that are common for anti-IL-17 mAbs include infectious complications, allergic reactions, hematological changes, liver enzyme elevation, etc. In contrast to anti-TNFα therapeutic means, IL-17 inhibitors showed a better safety profile regarding tuberculosis. For example, secukinumab does not increase the risk of tuberculosis [176]; only a few studies showed that single patients developed latent tuberculous infections (LTBI), but active tuberculosis was not registered [177]. Moreover, this class of anti-IL-17 drugs showed no influence on the development and progression of demyelinating diseases and systemic lupus erythematosus [178].

As we already mentioned in Section 3, the use of anti-IL-17 mAb is associated with IBD development and changes in gut microbiota. The majority of cases of IBD development were described for secukinumab, and lower occurrences were reported for ixekizumab, probably due to a longer therapeutic experience for secukinumab [179]. Brodalumab is only registered for psoriasis treatment, but the administration manual points to active Crohn’s disease as a contraindication. In studies with bimekizumab, low frequency of IBD development was noted [180]; nevertheless, according to the EMA, it should not be used in patients with IBD (https://www.ema.europa.eu/en/documents/product-information/bimzelx-epar-product-information_en.pdf, accessed on 29 April 2023).

Among the side effects of therapeutic antibodies, special attention is paid to psychiatric disorders and suicidal intents, which were shown for belimumab and some anti-TNF drugs [181]. Brodalumab takes special place in the anti-IL-17 group with regards to suicidal behavior. In 2015, a phase III clinical trial was stopped ahead of time due to suicide in six participants [182]. In 2017 brodalumab was approved for use, but highest priority has to be given for patient’s psychiatric status and the evaluation of the possibility for applying the treatment [181].

It should be noted that therapeutic schemes involving monoclonal antibodies for IL-17 inhibition have their own limitations. One of them is the immunogenicity of antibodies that results mainly in the formation of antidrug antibodies [33]. This phenomenon can impact the bioavailability, pharmacokinetic, and pharmacodynamic properties of the drugs, thus bringing a decrease in efficacy, and sometimes adverse reactions to therapy [183]. An exact mechanism of immunogenicity development during antibodies-based therapy has not yet been fully studied [184]. The analysis of the correlation between the pathogenesis of a specific rheumatological nosology and the appearance of antidrug antibodies revealed a higher probability of immunogenicity in patients with rheumatoid arthritis than in patients with spondyloarthritis [185]. Similarly, Benucci et al. [186] showed that patients with HLA-DRβ-11, HLA-DQ-03, and HLA-DQ-05 alleles had a higher risk of developing antidrug antibodies. Another study found an association between the HLA-DQ-05 genotype and the formation of antibodies to infliximab and adalimumab [187]. Moreover, the immunogenicity level depends on the particular IL-17A inhibitor used for therapy. For instance, Bagel et al. found anti-drug antibodies during brodalumab therapy in 2.7% of cases [188]. At the same time, Gordon et al. registered the immunogenicity effect in 9% of cases when using ixekizumab in patients with skin psoriasis [156]. Repeated administration of secukinumab led to immunogenicity development in 0.41% of cases [189].

Another limitation for the clinical application of monoclonal antibodies is their high sensitivity to storage and transportation conditions. Changes in temperature or humidity can affect the protein structure, making it ineffective or increasing its immunogenicity [190]. Moreover, production of monoclonal antibodies is a rather time-consuming and high-cost procedure [191].

Although not crucial, the abovementioned limitations of monoclonal antibodies have stimulated the search for alternative targeted IL-17 inhibitors with less immunogenicity, better storage and transportation stability, ease of application, and higher economic efficiency.

### 4.2. Small-Molecule Inhibitors

One of the strategies of IL-17 suppression is the use of inhibitors of certain molecular targets involved in IL-17 production and regulation. Small-molecule inhibitors specific to certain key enzymes involved in regulatory cascades seem to be very promising due to their simple structure, targeted immunomodulating action, and the possibility of oral administration. Among them, inhibitors of Janus kinases (JAKs) attract great attention. A number of JAK inhibitors have been approved for the treatment of inflammatory diseases including axSpA, PsA, and ulcerative colitis [192]. This group of targeted synthetic molecules block JAK1, JAK2, JAK3, and TYK2 in human cells and modulate the activity of different proinflammatory mediators, in contrast to monoclonal antibodies inhibiting only one specific cytokine. The efficacy of JAK inhibitors for SpA treatment was confirmed in different clinical trials (Table 3) [193,194,195]. Upadacitinib and tofacitinib are approved for axSpA treatment; tofacitinib, upadacitinib, and filgotinib are approved for PsA; and baricitinib is approved only for rheumatoid arthritis. Moreover, filgotinib demonstrated inhibition of inflammation associated with axSpA. Currently, the results of a phase II clinical trial showed the sufficient efficacy and safety of filgotinib in axSpA treatment. However, it has not been officially approved for axSpA so far [196]. However, among the unwanted effects of JAK inhibitor treatment, the most serious are cardiovascular and thromboembolic events [197]. Moreover, JAK inhibitor therapy can increase the risk of a Herpes zoster infection in rheumatological patients [198]. An FDA-sanctioned post-marketing Oral Rheumatoid Arthritis Trial (ORAL) Surveillance comparing the JAK inhibitor tofacitinib with anti-TNF therapy in patients with rheumatoid arthritis aged 50 years or older with at least one cardiovascular risk factor demonstrated a higher incidence of cardiovascular and oncological diseases, as well as different forms of herpes infection, when using tofacitinib in patients 65 years of age and older, while the drugs showed equivalent efficacy [199]. ASAS/EULAR recommends avoiding JAK inhibitor treatment in patients above the age of 50 with one or more additional cardiovascular risk factors and to those above the age of 65 [32].

Apart from JAK inhibitors, other small-molecule drugs of the same type also show good potential for blocking IL-17. In particular, inhibition of Rho-associated kinase (ROCK) by a specific inhibitor KD025 (SLx-2119, belumosudil) was used to suppress the pro-inflammatory T-cell response. A clinical study involving 38 patients with psoriasis vulgaris demonstrated a significant decrease in IL-17 concentration, while the concentrations of IL-6 and TNFα in serum have not changed significantly [209].

The differentiation of Th17 cells involves an expression of the nuclear gamma-t receptor (RORγt), which is induced upon the stimulation of the CD4+ by TGF-β and IL-6 [210]. In mice models, the transfer of T-cells with a genetic deficiency of RORγt failed to induce autoimmune encephalomyelitis and colitis; RORγt-null T-cells also showed profound defects in Th17 differentiation and production of cytokines [211]. The small-molecule inhibitors of RORγt also show promise for suppressing the IL17/IL-17R axis. The studies of Guendisch et al. [212] on Lewis rats with antigen-induced arthritis showed the efficacy of RORγt inhibition by a small-molecule compound Cpd 1, which suppressed the differentiation of Th17 cells and alleviated arthritis manifestations. Venken et al. [213] found that BIX119, an RORγt agonist/inhibitor, blocks IL-17A production in innate-like T-cells from SpA patients. Xue et al. [214] reported preclinical and phase I clinical studies of JNJ-61803534, an inverse agonist of RORγt. The compound provided a dose-dependent decrease in ex vivo-stimulated IL-17A levels in the blood, significantly reduced inflammation in mice collagen-induced arthritis and psoriasis models, and showed an acceptable clinical safety profile.

Clinical trial of the LY3509754 molecule (Eli Lilly and Company, Indianapolis, IN, USA), a small-molecule oral IL-17A inhibitor, was stopped in phase I due to serious side effects (clinicaltrials.gov Identifier: NCT04586920). Otherwise, DC-806 from DICE Therapeutics, Inc., an oral inhibitor of IL-17A that blocks its interaction with the cognate receptor, has been approved for a phase I clinical trial for psoriasis treatment. The small molecule inhibitor DC–853 (DICE Therapeutics, Inc., South San Francisco, CA, USA) and oral IL-17A inhibitor C4XD (C4X Discovery and Sanofi, Paris, France) are now undergoing preclinical studies.

Therefore, application of small-molecule inhibitors for SpA treatment is a prospective approach owing to oral administration, modulation of immune system functions, and high clinical efficiency. However, serious risks of cardiovascular and infectious events significantly limit their use in an elderly age group of patients.

### 4.3. Therapeutic Nucleic Acids

One of the most prospective alternatives to monoclonal antibodies could be therapeutic nucleic acids. Nucleic acids can be chemically synthesized using automated synthesizers; they are stable in a wide temperature range and tolerant to a wide range of storage and transportation conditions. Due to their relatively small size, nucleic acids are rarely immunogenic and usually do not cause any significant side effects. The most commonly used therapeutic nucleic acids are aptamers, siRNA, shRNA, and antisense oligonucleotides. Aptamers are short single-stranded RNA or RNA fragments that could specifically bind their targets due to the formation of a unique spatial structure. They are also called chemical antibodies due to their antibody-like abilities to bind their targets with high affinity and selectivity, and to directly inhibit the functional activity of the target. Up to now, a large number of aptamers have been selected against different types of targets, from small molecules or peptides to proteins and live cells and organisms. Among this diversity, there are aptamers specific to different cytokines [215] and their receptors, including IL-17A and IL-17RA (see Table 4).

An IL-17A-specific 2’-fluoro-modified RNA aptamer Apt21-2 was successfully used for blocking IL-17A interaction with IL-17RA in vitro [216]. The aptamer was also tested in model animals and provided an improvement in symptoms in rheumatoid arthritis and experimental autoimmune encephalomyelitis mouse models. Apt21-2 was also evaluated in a co-culture system mimicking psoriatic inflammation, using T-cells isolated from healthy donors and psoriatic patients [223]. The aptamer showed a neutralizing effect, which resulted in a decrease in IL-6, IL-8, and MCP-1 levels. However, in primary human keratinocytes, Apt21-2 did not affect proinflammatory cytokines levels. Authors showed a nonspecific RNA uptake by keratinocytes that could cause low Apt21-2 concentration in extracellular medium and a lack of neutralizing effect.

S. Hekmatimoghaddam et al. studied the therapeutic potential of gelatin hydrogel supplied with cerium oxide nanoparticles coated by an IL-17-specific aptamer [224]. Treatment with the obtained nanocomposite caused a significant decrease in IL-17 concentration in the serum of model animals with induced brain inflammation.

Adachi et al. produced a highly selective 2’-fluoro-modified RNA aptamer AptAF42dope1 which only discriminated the IL-17A/F heterodimer [217]. This aptamer decreased production of GRO-α in cell culture activated by IL-17A/F, but its therapeutic potential should be further investigated.

Other DNA aptamers, M2 and M7 specific to IL-17A, were successfully used in the mouse model of psoriasis induced by imiquimod [218]. Topical administration of aptamers resulted in a decrease in keratinocyte hyperproliferation and inflammatory response, and downregulated IL-17A production.

Chen et al. selected the IL-17RA-specific DNA aptamer RA10-6 using live cell culture as a target [219]. The obtained aptamer blocked IL-17/IL-17RA interaction in a dose-dependent manner in a mouse model of osteoarthritis. A histological study of the synovial membrane showed a decrease in its thickening and a decrease in the concentration of IL-16 in the synovium.

Aside from direct blocking of the IL-17A/IL-17RA pathway, various therapeutic nucleic acids could downregulate the IL-17A/17RA axis at the mRNA level. The main challenge of this strategy is the targeted delivery of such nucleic acids into IL-17A-producing Th17 cells. It is well known that primary Th17 cells are difficult to transfect using conventional methods. Moreover, RNA interference machinery does not work in T-cells as efficiently as in other cell types [225]. However, there are some prospective strategies for nucleic acid delivery into Th17 cells based on using nanoparticles and aptamers as transporters. For instance, a CD4-specific RNA aptamer provided targeted delivery of RORγt shRNA into CD4+ cells [221]. Simultaneously, a CD30-binding aptamer was conjugated to RORγt shRNA for IL-17 downregulation [222]. RORγt (retinoic acid-related orphan receptor γt) is the master transcription factor that controls differentiation of Th17 cells and synthesis of Th17 cytokines. An inhibition of RORγt at the mRNA level provides specific suppression of Th17 cells. Chimeric aptamer-shRNA constructs successfully inhibited the RORγt receptor, resulting in a decrease in IL-17A and IL-17F production.

Liu et al. used liposomes for the delivery of antisense oligonucleotides targeted to IL-17RA [220]. Topical treatment with oligonucleotide–liposome complexes caused clinical and histological improvement in the imiquimod-induced psoriasis mouse model. According to quantitative real-time PCR and western blotting analyses, such an improvement was associated with a decrease in IL-17RA mRNA level. Antisense oligonucleotides complexed with liposomes were also evaluated in psoriasis-like human skin models generated by treatment of 3D organotypic rafts with a cytokine mix. This experiment demonstrated a significant decrease in IL-17RA mRNA level.

Clearly, the treatment protocols for therapeutic nucleic acids (effective dose, way of administration, additional chemical modifications, etc.) should be further improved for possible clinical applications. Therefore, available preliminary data allow for the consideration of therapeutic nucleic acids as very promising agents for the treatment of rheumatic pathologies.

### 4.4. Other Aproaches to IL-17 Downregulation

Another class of potential therapeutics for suppressing IL-17 are linear peptides. These compounds act as antagonists of IL-17A and could block its interaction with the receptor. Liu et al. studied in detail the mechanism of action of these peptides and estimated their therapeutic potential [226].

Vaccination against IL-17A is a prospective way to produce patients’ own antibodies against IL-17A. A virus-based vaccine Qβ-IL-17 was used for the immunization of model animals with an autoimmune disease [227]. Mouse vaccinations resulted in a decrease in disease manifestation.

An affibody Izokibep (ACELYRIN, Inc., Affibody, Inmagene Bio, Agoura Hills, CA, USA), and an IL-17A-specific protein that belongs to so-called antibody mimics, provides one more alternative to traditional monoclonal antibodies. According to developers, the protein shows a 1000-fold higher target binding affinity than a typical monoclonal antibody, together with a much smaller size (about 1/8 of an antibody). Izokibep provided good efficacy in a phase II clinical trial (135 participants) [228]: by the 16th week, 52% of the patients on an 80 mg dosage and 48% on a 40 mg dosage reached an ACR50 response. No severe adverse effects were registered, although mild infections (i.e., candidosis) and erythema were registered. Recently, a phase II clinical trial (25 participants) was completed for izokibep in SpA treatment, but the results are yet to be published (clinicaltrials.gov Identifier: NCT04795141).

## 5. Conclusions and Future Prospects

Spondyloarthritis consolidates a group of chronically progressing autoimmune diseases of the spine and peripheral joints characterized by a high prevalence in the population, significant social and economic loss, early disability, and high costs for treatment and rehabilitation. SpA pathogenesis greatly depends on the imbalance between pro- and anti-inflammatory cytokines, with a hyperproduction of IL-17, IL-23, and TNFα.

At the moment, the main tools to control chronic inflammation are represented by pharmacological agents that suppress the level of pro-inflammatory cytokines in a targeted manner, namely monoclonal antibodies against the abovementioned cytokines and Janus kinase inhibitors. Among all the approaches, only the inhibition of IL-17 activity allows for controlling both clinical manifestations (arthritis, enthesitis, spondylitis, and skin lesions) and spinal structural changes after a mere two years of treatment. Despite the obvious success of this approach, current means of IL-17 axis suppression are complicated by severe side effects, such as the activation of a latent tuberculosis infection and secondary inefficacy due to the immunogenicity of the drug. It is also necessary to note a high level of direct medical costs for targeted therapy by inhibitors of IL-17 or other pro-inflammatory cytokines. Therefore, the search for alternative strategies and approaches remains a very important task in this field of biomedicine. Among them, we should make a point of small molecule inhibitors of targets other than JAKs, therapeutic nucleic acids (aptamers, shRNAs and antisense oligonucleotides), and affibodies. The completely different chemical nature and mechanism of biological action of these molecules might provide a possibility for optimizing the safety profile and cost of the therapy. It should be emphasized that novel dosage forms of these IL-17 axis inhibitors might potentially eliminate the logistic problems of cold chain transportation, solve the issue of clinically relevant immunogenicity, and minimize the extent of patients’ dependence on specialized medical centers.

Meanwhile, the key role of IL-17 is also characteristic for non-rheumatic nosologies. For instance, hidradenitis suppurativa (HS) represents a chronic skin disease accompanied by the forming of nodules, abscesses, and fistulas. The intensity of skin lesions deeply affects the psycho-emotional state of patients and decreases life quality [229]. A number of studies have shown elevated levels of IL-17 and IL-23 in the skin of HS patients [230]. The studies of IL-17 inhibitors for HS treatment are now actively ongoing; ixekizumab, secukinumab, and brodalumab provided an improvement in skin lesions in a part of patients [231]. An important role of the IL-17 axis has also been shown for Behçet’s syndrome. The main clinical manifestations include skin and mucous lesions and severe uveitis which leads to blindness. Elevated levels of IL-17 are also associated with skin and mucus lesions, maintenance of high disease activity, and development of uveitis [232]. Preliminary studies demonstrated the efficacy of the secukinumab for patients with Behçet’s syndrome [233]. Moreover, the role of the IL23/IL17 axis was also proved for eye disorders in other pathologies, such as sympathetic ophthalmia, eye lesions at sarcoidosis, and chorioretinitis [234]. IL-17A is now considered an important player in the pathogenesis of human respiratory diseases such as asthma, chronic obstructive pulmonary, and cystic fibrosis. Targeted suppression of the IL-17/IL17R axis appears to be a promising method of treatment [235,236,237].

An established significance of IL-17 hyperproduction in the pathogenesis of a row of socially important diseases, and the long-term positive experience of the use of anti-IL17A antibodies in the treatment of spondyloarthritis and psoriasis, provide a strong basis for broadening the spectrum of therapeutic indications for the strategy of IL-17 suppression for non-rheumatic pathologies. A state-of-the-art in the field of development of targeted strategies of IL-17 inhibition would bring us novel tools for implementing this strategy in the foreseeable future.

## Figures and Tables

**Figure 1 biomedicines-11-01328-f001:**
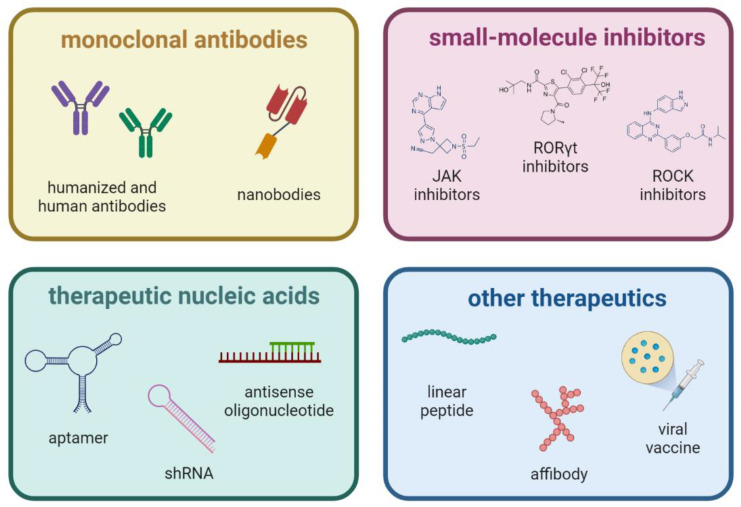
Types of molecules for targeted action on IL-17 axis.

**Figure 2 biomedicines-11-01328-f002:**
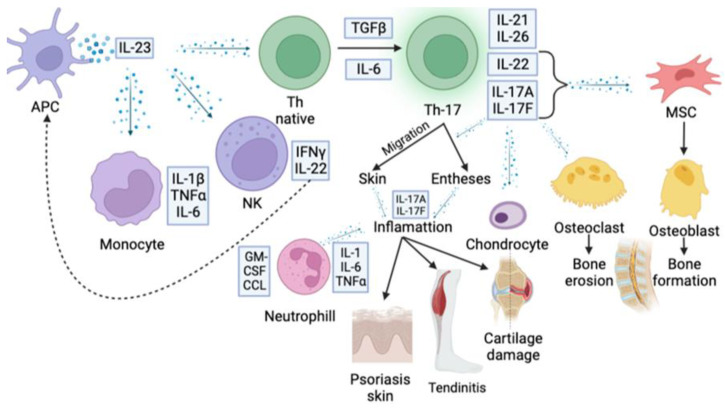
Immune cells and cascade of cytokines involved in SpA pathogenesis.

**Table 1 biomedicines-11-01328-t001:** Physiological and pathological effects of IL-17.

IL-17 Family Members	Physiological Effects	Pathological Effects
IL-17A	Bone remodeling (stimulation of osteoclastogenesis) [45] Recruitment of myeloid cells to the site of infection [46] Participation in immune responses against extracellular fungal [47] and bacterial agents [48] Maintenance of intestinal microbiota homeostasis [49] Maintenance of the epithelial barrier in the gut [50] Promotes tissue healing by activation of proliferation [51] Induces the production of bradykinin in the epithelial cells of the tubules of the kidneys in acute injury [52] Participation in the regulation of glucose and lipid metabolism [53]	Development of inflammatory arthritis [35] Possible link between depression and increased levels of IL-17 [54] Key role in the pathogenesis of spondyloarthritis [55] Injurious role in ischemic stroke [56] Involvement in the progression of neurocognitive disorders [57] The key link in the pathogenesis of psoriasis [58] The role in the development of liver fibrosis is being studied [59]
IL-17B	Blocks IL-25 signaling [60] TNF-α production induction [61] Participates in the process of embryonic development of bone tissue [62] Participates in the healing of bone fractures [63] The role in the regeneration of liver cells is being studied [64]	Key role in the progression of tumors: gastric [65], pancreas, lungs and breast [66], incl. increases the risk of metastasis [67] Activation and maintenance of chronic inflammation [68] Involved in the development of inflammatory arthritis [69] The role in the pathogenesis of systemic lupus erythematosus is being studied [70]
IL-17C	Participation in the regulation of the innate immune response in epithelial cells [71] Protection of the peripheral nervous system during the activation of the herpes virus [72]	Development of psoriatic skin lesions [73] Participation in the development of skin lesions in atopic dermatitis [74] Aggravation of the course of autoimmune encephalitis [75] The role in the development of kidney damage in SLE is being studied [76]
IL-17D	Antitumor immune response [77] Probably involved in local immune reactions, inhibition of hematopoiesis [78] Antiviral immune response in cytomegalovirus infection [79] Regulation of homeostasis in the intestine, probably anti-inflammatory effect in colitis [80] Possible role in the development of the immune response in bacterial infection [81]	Possibly involved in the development of severe sepsis [82]
IL-17E (IL-25)	Participation in the immune response to parasitic invasion [83,84,85] Anti-inflammatory effects in the colonic mucosa [86], while there is evidence of pro-inflammatory activity in the colonic mucosa [60] Anti-inflammatory activity in the central nervous system [87] Participation in the development of thymus cells [88]	Participation in the development of psoriasis, however, no convincing evidence for its role in the development of articular syndrome has been obtained [89] Induction of inflammatory reactions of the allergic type [90] Exacerbation of bronchial asthma [91]
IL-17F	Immune reactions in mucous membranes, including antifungal immune response [92] Combined effects with IL-17A.	The role in the development of psoriasis and psoriatic arthritis is being studied [93] Activation of mucin hypersecretion, participation in inflammation in bronchial asthma [94]

**Table 2 biomedicines-11-01328-t002:** Anti-IL-17 therapeutic monoclonal antibodies.

Monoclonal Antibody	IL-17 Family Member	Therapeutic Indications	State Registration
Secukinumab	IL-17A	Plaque psoriasis Psoriatic arthritis Radiographic axial spondyloarthritis Non-radiographic axial spondyloarthritis Enthesitis-related arthritis	Food and Drug Administration USA (FDA US)—2015 European Medicines Agency (EMA)—2015 State Register of Medical Remedies Russian Federation (SRMR RF)—2016
Ixekizumab	IL-17A	Plaque psoriasis Psoriatic arthritis Radiographic axial spondyloarthritis Non-radiographic axial spondyloarthritis	FDA US—2016 EMA—2016 SRMR RF—2018
Netakimab	IL-17A	Plaque psoriasis Radiographic axial spondyloarthritis Psoriatic arthritis	SRMR RF—2019
Bimekizumab	IL-17A IL-17F	Plaque psoriasis	EMA—2021
Brodalumab	IL-17A-receptor	Plaque psoriasis	FDA US—2017 EMA—2017

**Table 3 biomedicines-11-01328-t003:** JAK inhibitors.

JAK Inhibitor	Main Selectivity to JAK Isoform	Therapeutic Indications	Trials in SpA	State Registration
Approved for spondyloarthritis				
Tofacitinib	JAK1, JAK2, JAK3	Rheumatoid arthritis Psoriatic arthritis Ulcerative colitis Radiographic-axial spondyloarthritis Active polyarticular juvenile idiopathic arthritis Juvenile psoriatic arthritis in patients 2 years of age and older Plaque psoriasis	Efficacy and safety of Tofacitinib in subjects with active rx-axSpA: phase III (NCT03502616) [200] Efficacy and safety of Tofacitinib in psoriatic arthritis: comparator study OPAL BROADEN: phase III (NCT01877668) [201] Tofacitinib in psoriatic arthritis subjects with inadequate response to TNF Inhibitors OPAL BEYOND: phase III (NCT01882439) [202]	FDA US—2012 EMA—2017 SRMR RF—2013
Upadacitinib	JAK1	Rheumatoid arthritis Psoriatic arthritis Ulcerative colitis Radiographic-axial spondyloarthritis Nonradiographic-axial spondyloarthritis	A study evaluating the safety and efficacy of upadacitinib in adults with active rx-axSpA SELECT-AXIS 1: phase II/III (NCT03178487) [203] A study to evaluate efficacy and safety of upadacitinib in adults with axSpA SELECT AXIS 2: phase III (NCT04169373) [196] A study comparing upadacitinib to placebo in participants with active psoriatic arthritis who have a history of inadequate response to at least one bDMARD SELECT-PsA 2 (NCT03104374) [204]	EMA—2019 FDA—2019 SRMR RF—2019
Baricitinib	JAK1, JAK2	Rheumatoid arthritis Atopic dermatitis Alopecia areata COVID-19	A randomized phase 2b trial of baricitinib, an oral Janus kinase (JAK) 1/JAK2 inhibitor, in patients with moderate-to-severe psoriasis [205], but clinical development of baricitinib for the treatment of PsA has been halted [206]	EMA—2016 FDA—2018 SRMR RF—2018
Filogotinib	JAK1	Rheumatoid arthritis Ulcerative colitis	A study to assess efficacy and safety of filgotinib in rx-axSpA TORTUGA: phase II (NCT03117270) [207] An open-label, long-term extension study with filgotinib in active psoriatic arthritis: phase II (NCT03320876)	EMA—2020
Not approved for rheumatological applications				
Deucravacitinib	TYK2	Plaque psoriasis	Efficacy and safety of BMS-986165 compared with placebo in participants with active psoriatic arthritis: phase II (NCT03881059) [208]	FDA—2022 EMA—2023

**Table 4 biomedicines-11-01328-t004:** Therapeutic nucleic acids for IL17A regulation.

Target	Therapeutic Nucleic Acid	Experimental Model	Ref.
IL-17A	2’-F-RNA aptamers Apt21-2 and Apt3-4	mouse model of multiple sclerosis and inflammatory arthritis	[216]
	2’-F-RNA aptamer AptAF42dope1	primary human foreskin fibroblast BJ cells	[217]
	DNA aptamers M2 and M7	imiquimod induced psoriasis mouse model	[218]
IL-17RA	DNA aptamer RA10-6	mouse model of osteoarthritis	[219]
	Liposomes + antisense oligonucleotide	imiquimod induced psoriasis mouse model human cytokine-induced psoriasis skin model	[220]
Th17 cells	CD4 aptamer + RORγt shRNA	CD4+ cells	[221]
	CD30 aptamer + RORγt shRNA	CD30+ and CD4+ cells	[222]

## Data Availability

Not applicable.

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
