# Peer review of "The Interleukine-17 Cytokine Family: Role in Development and Progression of Spondyloarthritis, Current and Potential Therapeutic Inhibitors"

_biomedicines, 2023, doi:10.3390/biomedicines11051328_

Round 1

Reviewer 1 Report

The authors comprehensively discuss the role of IL-17 in spondylarthritis. However, the manuscript needs some clarification and elaboration:

Remarks:

1) The chapter on JAK inhibitors should be placed after the chapter on anti-IL-17A/anti-IL17RA monoclonal antibodies since JAKinhibitors are a second-choice therapy.

2) The Oral surveillance study should be mentioned in the text when discussing concerns about the safety of JAKinhibitors.

3) It should be better clarified that the term ankylosing spondylitis is now generally replaced by the term axial radiographic spondylitis (rx-axSpA).

4) It is necessary to specify the characteristics of nonradiographic axial spondylitis (nr-axSpA) such as the presence of bone edema at the sacroiliac joints visible by MRI imaging.

5) It should be specified that netakimab differently from secukinumab and ixekizumab has not been approved by FDA and EMA but only in Russia for the treatment of axSpA.

 6) The role of IL-23 and IL-17 in the differentiation and stabilization of the phenotype of Th17 cells should be better detailed. The concept of "IL-23-IL17 axis" should also be explained.

7) Table 2 should be clearer in indicating that tofacitinib (rx-axSpA) and upadacitinib (rx-axSpA and nr-axSpA) have been approved for axSpa while other JAKs already available for clinical use in rheumatology such as baricitinib or filgotinib have no such approval. Add to that section the citation PMID: 36674537.

8) The mechanism of flexibility of Th17 RORgt+ phenotype that can transform into FOXP3+ Treg phenotype in the absence of IL-23 and vice versa should be described briefly.

9) The role of IL-17 in juvenile arthritis (JIA)-equivalent forms of adult axSPA should be mentioned and cited (PMID: 36363508).

English needs minor editing

Author Response

Dear Editors,

We are grateful to Biomedicines journal and the reviewers for interest to our work and for the critical remarks and suggestions which encouraged us to improve the initial manuscript. We thoroughly revised the manuscript according to reviewers’ remarks. All changes are marked by blue color.

The point-to point answers to reviewers’ remarks are listed below.

Thank you for the attention to our manuscript.

Reviewer’s #1 remarks:

  • The chapter on JAK inhibitors should be placed after the chapter on anti-IL-17A/anti-IL17RA monoclonal antibodies since JAK inhibitors are a second-choice therapy.

We agree with this remark about JAK inhibitors being a second-choice therapy. We placed the JAK inhibitors chapter after chapter on monoclonal antibodies.

  • The Oral surveillance study should be mentioned in the text when discussing concerns about the safety of JAK inhibitors.

We agree with this remark, so we added a fragment discussing ORAL surveillance study into the manuscript (page 13, line 543).

  • It should be better clarified that the term ankylosing spondylitis is now generally replaced by the term axial radiographic spondylitis (rx-axSpA).

We agree with this remark. We substituted term ankylosing spondylitis with term radiographic axial spondylitis in the manuscript.

  • It is necessary to specify the characteristics of nonradiographic axial spondylitis (nr-axSpA) such as the presence of bone edema at the sacroiliac joints visible by MRI imaging.

We agree with this remark. So, we added a new paragraph describing the key characteristics of nr-axSpA into the manuscript (see page 6, line 192).

  • It should be specified that netakimab differently from secukinumab and ixekizumab has not been approved by FDA and EMA but only in Russia for the treatment of axSpA.

Indeed, netakimab is approved for treatment of rx-axSpA in Russian Federation and Belarus and hasn’t been yet approved by FDA and EMA. We added this information into the manuscript (see page 11, line 414).

  • The role of IL-23 and IL-17 in the differentiation and stabilization of the phenotype of Th17 cells should be better detailed. The concept of "IL-23-IL17 axis" should also be explained.

We agree that IL-23 and IL-17 play a critical role in the differentiation and stabilization of Th17 cells phenotype. We added relevant chapter 2 “IL23/IL17 axis in the pathogenesis of autoimmune inflammation” into revised manuscript (see page 3).

  • Table 2 should be clearer in indicating that tofacitinib (rx-axSpA) and upadacitinib (rx-axSpA and nr-axSpA) have been approved for axSpa while other JAKs already available for clinical use in rheumatology such as baricitinib or filgotinib have no such approval. Add to that section the citation PMID: 36674537.

Table 2 was edited according to reviewer’s suggestion (see page 14). We added subheads to indicate approval of JAK inhibitors for axSpA treatment. The proposed reference (PMID 36674537) was cited in (see page 13, line 540).

  • The mechanism of flexibility of Th17 RORgt+ phenotype that can transform into FOXP3+ Treg phenotype in the absence of IL-23 and vice versa should be described briefly.

We added a new fragment discussing the flexibility of Th17 RORgt+ phenotype into the revised manuscript (see chapter 2, page 4, line 139).

  • The role of IL-17 in juvenile arthritis (JIA)-equivalent forms of adult axSPA should be mentioned and cited (PMID: 36363508).

A requisite subsection on juvenile arthritis was added into the manuscript (see page 9, line 318). The proposed reference (PMID 36363508) was cited (see page 13, line 327)

Reviewer 2 Report

In this review, ,, The Interleukine-17 Cytokine Family: Role in Development and Progression of Spondyloarthritis, Current and Potential Therapeutic Inhibitors,, by Anna Davydova et al. , the authors summarized literature data on the role of IL-17 family in the pathogenesis of spondyloarthritis and analyzes existing therapeutic strategies for IL-17 suppression with monoclonal antibodies and Janus kinase inhibitors.

The article has some shortcomings:

-   I made some suggested changes in the body of the manuscript;

-   double-check abbreviations and make the necessary corrections so that abbreviations are explained when they first appear, both in the abstract and in the manuscript text and figure legends.

Overall Recommendation: Accept after minor revision.

Minor editing of English language required

Author Response

Dear Editors,

We are grateful to Biomedicines journal and the reviewers for interest to our work and for the critical remarks and suggestions which encouraged us to improve the initial manuscript. We thoroughly revised the manuscript according to reviewers’ remarks. All changes are marked by blue color.

The point-to point answers to reviewers’ remarks are listed below.

Thank you for the attention to our manuscript.

Reviewer’s #2 remark:

double-check abbreviations and make the necessary corrections so that abbreviations are explained when they first appear, both in the abstract and in the manuscript text and figure legends

We have thoroughly revised the whole manuscript and explained all abbreviations. We also added a list of abbreviations at the end of the manuscript.

Round 2

Reviewer 1 Report

The authors fully answered my questions. 

English needs only minor changes.